
# Wave climate and storm activity in the Kara sea

Stanislav Myslenkov[1,2,3], Vladimir Platonov[1], Alexander Kislov[1], Ksenia Silvestrova[2], Igor Medvedev[2,4]

[1]Lomonosov Moscow State University, 119991, Moscow, Russia

[2]Shirshov Institute of Oceanology RAS, 117997, Moscow, Russia

[3]Hydrometeorological Research Centre of the Russian Federation, 123242, Moscow, Russia

[4]Fedorov Institute of Applied Geophysics, 129128, Moscow, Russia

*Correspondence to*: Stanislav Myslenkov (stasocean@gmail.com)

**Abstract.** Recurrence of extreme wind waves in the Kara Sea strongly influences the Arctic climate change. The paper presents the analysis of wave climate and storm activity in the Kara Sea based on the results of numerical modeling. A third-generation wave model WaveWatchIII is used to reconstruct wind wave fields on an unstructured grid with a spatial resolution of 15–20 km for the period from 1979 to 2017.

The mean and maximum wave heights, wavelengths and periods are calculated. The maximum significant wave height (SWH) for the whole period amounts to 9.9 m. The average long-term SWH for the ice-free period does not exceed 1.3 m. The seasonal variability of the wave parameters is analyzed.

The interannual variability of storm waves recurrence with different thresholds (from 3 to 7 m) was calculated. A significant linear trend shows an increase in the storm wave frequency for the period from 1979 to 2017. A double growth in the reccurence was observed for cases with an SWH more than 3–5 m from 1979 to 2017. The local maximum of the storm waves more than 3–4 m was observed in 1995, and the minimum in 1998. The maximum value (four cases) of the number of storms with an SWH threshold 7 m is registered in 2016. The frequency of wind speeds and ice conditions contributing to the storm waves formation were analyzed. It is shown that trends in the storm activity of the Kara Sea are primarily regulated by the ice. If the ice cover decreases in the southern part of the sea that leads to the increase of the number of events only with SWH threshold more than 3–4 m. If in the entire sea the ice cover decreases that leads already to increase of the extreme storms. The frequency of strong and long-term winds has high interannual variability and a weak positive trend.

The analysis of distribution functions of the storm events with an SWH more than 3 m was carried out. Six different sectors of the Kara Sea were analyzed to reveal spatial differences. A comparison of the different distribution laws showed that the Pareto distribution is in the best agreement with the data. Up to 99% of the points are described by this distribution. However, the extreme events with an SWH more than 6–7 m deviate from the distribution, and their probability is approximately twice as less as that predicted by the Pareto distribution. Presumably, this deviation is caused by the combined impact of rare wind speed frequencies and anomalies of the sea ice conditions.

Keywords: Kara Sea, wave climate, storm activity, wind waves, wave modeling, WAVEWATCH III, probabilistic analysis, extreme waves

## 1. Introduction

The interest increases in the study of the hydrometeorological conditions of the Arctic seas due to the active economic development of this region. The active oil and mineral field exploration and development occur here, in this region. The Arctic is an area of intensive shipping and fishery. Wind, sea ice and wave conditions are limiting factors for the economic activity and the development of the infrastructure in the coastal zone. The storm waves can destruct the infrastructure facilities in coastal zones and offshore, a threaten human lifes and cause economic damage.





The construction of new facilities and the operation of existing ones need the risk and hazard assessment associated
with impact of the storm waves. We need to study the extreme winds and waves in the past, their interannual variability because
it possible to reduce the disasters risk in future.
Nowadays, storm activity is studied with several methods with use of different sources: direct observation data (De Leo et al.,
2020; Menéndez et al., 2008), altimetry data (Meucci et al., 2020; Young and Ribal, 2019; Liu et al., 2016) and modeling data
(Bertin et al., 2013; Dobrynin et al., 2015; Kurmar et al., 2016; Semedo et al., 2011; Wang and Swail., 2001; Weisse et al.,
2005). There is also research work on wave heights in the 21st century Arctic Ocean (Khon et al., 2011). As direct
measurements, especially in the Arctic Region, are very rare, and altimetry data are short series, thus the simulated data from
models are preferable.
Regular and extreme characteristics of wind and waves of the Kara Sea are given in the Wind and Wave Climate Handbook
of Russian Maritime Register of Shipping (2009). These data are based on the results of modeling. But the input wind forcing
for the simulations was calculated from the atmospheric pressure data. Subsequently it was verified and calibrated by the
weather stations measurements. Such information needs to be refined with the modern atmospheric reanalyses data. Diansky
et al. (2014) describes some new results devoted to wave hindcast and forecast of the Kara Sea using the WRF wind.
Stopa et al. (2016) showed the main features of the wave climate and trends in the whole Arctic for the period 1992–
2014. They noted that the ice cover decreases and at the same time the wave height rises. Liu et al. (2016) used satellite
observations (1996–2015) for studying the wave climatology in the Arctic Ocean in summer (August–September). They show
that winds and waves in the Barents and Kara Seas initially increased from 1996 to 2006 and later decreased until 2015.
Li et al. (2019) present details of the significant wave height (SWH) change with the retreat of the ice edge. The
increase of the wave heights is shown for the Arctic subregions, including the Kara Sea.
Interannual variations of the mean and extreme SWH in ice-free conditions in the Kara Sea are described in (Duan et
al., 2019). They estimated linear trends of SWH from 2005 to 2018, but these trends are not statistically significant for the
most areas. The mean and extreme SWHs show relatively positive trends in the northeastern part of the Kara Sea, but the
analysed period is too short for trend estimation (Duan et al., 2019).
Positive trends of the highest SWH and wind speed are shown for the Laptev and the Beaufort Seas based on the 38-
year-long reanalysis. But for the Kara Sea trend analysis was not realised (Waseda et al., 2018).
The wind wave characteristics are studied in several researches for the whole world ocean (Young et al., 2011;
Semedo et al., 2011; Kurmar et al., 2016). Semedo et al. (2011) described the seasonal variations of the global wave heights
from 1957 to 2002 with the ERA-40 reanalysis data. In the Barents Sea the positive linear trend of SWH in winter months is
observed Semedo et al. (2011). Zieger et al. (2013) calculated the mean and 99th SWH percentile for March and September in
the Arctic based on the Envisat satellite data from 2002 to 2012.
However, in all mentioned studies there is no deep analysis of the storm interannual variability in the Kara Sea or the
data series are too short to carry out it.
In (Kislov and Matveeva, 2020) studied the seasonal and interannual features of the Kara Sea meteorological regime
and its connection with circulation indices. The period 2000-2010 is characterized by significant climate warming, a reduction
of the surface of the old and first-year sea ice in the Arctic (Serreze et al., 2015; Caian et al., 2018; Shalina, 2013) and the
appearance of a significantly larger ice-free sea surface than earlier. These highlights with the changing thermobaric structure
of the atmosphere (Semenov et al., 2015; Semenov et al., 2018), and variability of Atlantic water inflow (Ivanov and Repina,
2018) can lead to the wind-wave regime changes in the Arctic Region. Such important features as modification of the cyclone
number and its trajectories (Tilinina et al., 2014; Zhang et al., 2004), and the increase in daily extremes of wind speed (Surkova
et al., 2015) were described. Also wind speeds rise to the north from 75–80° N in recent decades according to climatic reports
(IPCC, 2013). Positive trends in average and extreme wind speeds in some parts of the Arctic Region are also noted in (Young
and Ribal, 2019).





In this research, we present the wave reanalysis of the Kara Sea with a high spatial and temporal resolution. The
regular and extreme wave characteristics were studied. The recurrence, trends and probability analysis of the storm waves in
the Kara Sea were estimated for the long period from 1979 to 2017.

**2. Data and Methods**

**2.1 Wave modeling**
One of the main approaches of studying the world ocean wave climate is the spectral wave modeling that allows to
create long-term reanalyses of wave parameters (Kurmar et al., 2016; Reistad et al., 2011; Semedo et al., 2011; Weisse et al.,

2005;).

Modern spectral wave models provide high-quality results which are in good agreement with direct wave
measurements. Correlation between model results and measurements data is usually 0.8–0.9, and the standard error is 0.3 m
(Li et al., 2019, Reistad et al., 2011; Stopa et al., 2016).
The wave characteristics in the Kara Sea were calculated by the spectral wave model WAVEWATCHIII (WWIII)
version 4.18 (Tolman, 2014). This model parameters considers wind speed, ice concentration, effects of the energy dissipation,
non-linear interactions and bottom friction. This model is based on a numerical solution of the equation of the spectral wave
energy balance:
$$\frac{\partial E(\omega, \theta, \vec{x}, t)}{\partial t} + \vec{V}(\omega, \theta)\nabla E = S(\omega, \theta, \vec{x}, t),\qquad(1)$$

where $\omega$ and $\theta$ are the frequency and the propagation direction of the spectral component of the wave energy;
$E(\omega, \theta, \vec{x}, t)$ is the two-dimensional spectrum of the wave energy at a point with vector coordinate $\vec{x}$ at time point $t$ ;
$\vec{V}(\omega, \theta)$ is the group velocity of the spectral components; $S(\omega, \theta, \vec{x}, t)$ is a function that describes the wave energy
sources and sinks, i.e., the transfer of the energy from the wind to the waves, nonlinear wave interactions, dissipation of the
energy through collapse of the crests at a great depth and in the coastal zone, friction against the bottom and ice, wave scattering
by ground relief forms, and reflection from the coastline and floating objects. The energy balance equation is integrated using
finite-difference schemes by the geographic grid and the spectrum of wave parameters.
In present study, the calculations were made with the ST1 scheme (Tolman, 2014). A Discrete Interaction
Approximation (DIA) model was used for the possible nonlinear interactions of the waves. The DIA is a standard
approximation for the calculation of nonlinear interactions in all modern wave models.
Influence of the sea ice on the wave development was considered by the IC0 scheme, where a grid point is considered as ice-
covered if ice concentration is >0.25. Thus, the exponential attenuation of wave energy adjusted for the sea ice concentration
at a given point was added.
In the shallow water, the increase in wave height as waves approach the shore and the related wave breaking after
waves reach the critical value of steepness were taken into consideration. The whitecapping effect taken into account in the
ST1 scheme. The standard JONSWAP scheme was used to take the bottom friction into account. The spectral resolution of
the model is 36 directions (Dq = 10°), the frequency range includes 36 intervals (from 0.03 to 0.843 Hz).
The calculations were performed using the unstructured grid, which consists of 16792 nodes (Fig. 1). The bathymetry
data were obtained from the ETOPO 1-minute bottom topography database (https://www.ngdc.noaa.gov/mgg/global/) and
detailed navigation maps. The grid covers the Barents and Kara seas, as well as the entire northern part of the Atlantic Ocean.
The spatial resolution varies from ~15 km for the Kara Sea to ~50 km for the northern part of the Atlantic Ocean. The North
Atlantic was included into the grid because of the swell propagating into the Barents and Kara seas, it was shown earlier in
(Myslenkov et al., 2015).



More detailed description of the model configuration, the main features of the experiments with the unstructured mesh

is presented in (Myslenkov et al., 2018; Myslenkov et al., 2019). Wind and sea ice concentration data for the wave modeling
were taken from the NCEP/CFSR reanalysis (1979–2010) with a spatial resolution ~ 0.3° (Saha et al., 2010) and NCEP/CFSv2
reanalysis (2011–2017) with a resolution ~ 0.2° (Saha et al., 2014), temporal resolution is 1 hour.
Figure 1

As a result, we got the wind wave fields for every three hours from 1979 to 2017 (total 39 years). The model results

include the SWH (average value from 1/3 of the highest waves), the wave propagation direction, the mean wave period (WP)
Tm02 and mean wave length (WL). Also, the wave heights of 1% and 3% probability of exceedance were used for the data
analysis. These values were calculated as 1.51×SWH and 1.32×SWH, respectively (Coastal Egineering, 1995). The maximum
and long-term SWH were calculated based on these data. When the Kara Sea was ice covered the wave parameters were equal
to zero in model results. The mean long-term characteristics were performed for the ice-free period when the wave parameters
were nonzero.

**2.2 Recurrence of the storm waves**

The storm activity analysis was held according to the Peak Over Threshold (POT) method, which is widely used (De

Leo et al., 2020; Menéndez et al., 2008). This method was previously used for the Barents Sea (Myslenkov et al., 2019). The
number of storm waves with different SWH from 3 to 7 m was calculated for each year in the Kara Sea or within the sea sector.
The calculation procedure includes the following steps: if at least one node in the investigated sea area has the SWH exceeding
a threshold, then such event is attributed to the storm case with waves more than this threshold. This event continues until the
SWH will not be less than the threshold at all nodes of the investigated area. Further, if the threshold is exceeded in one of the
nodes again, then this event is added to the following case. A period of 9 hours at least should pass between two storm cases
for eliminating the possible errors. This technique has an inaccuracy associated with storms running in a row or from different
directions at the same time. However, such cases are rare. The proposed algorithm works correctly, it was validated by a visual
analysis conducted for several years.

The 3 m threshold was chosen as the 99% percentile of the entire studied period (1979–2017 in 3-hour interval) for

the points in the central part of the sea (including the ice-covered periods) or as the 95% percentile for the ice-free period.
There could be different criteria for the 95th percentile for regions of the Kara Sea. However, the aim of this research is  deep
analysis of the extreme events with the SWH exceeding 3 m for any points of the Kara Sea.

**2.3 Quality assessment of the wave model results**

Quality assessments of the modeling results based on the instrumental wave measurement data of the 1 mooring

station in the Kara Sea (Fig. 2) for the period october 2012, published in [Atlas ..., 2015]. Wind wave measurements were
carried out with upward-looking sonar IPS-5 for ice profiling. Data from this Atlas was digitized for statistical analysis.
Figure 2

A comparison of the modeled and measured SWH from September 1 to October 22, 2012 for mooring station is shown

in Fig. 3. The model provides the absolute wave height and the phase of the individual storm event quite well. The result of
the comparison for the entire data array is shown in Fig. 4. The correlation coefficient is 0.91, the BIAS is 0.08 m, and the
RMSE is 0.31 m. The Scatter Index is 0.28. Further in the analysis the SWH values are presented with an accuracy of one
decimal place during further analysis due to the obtained quality estimates.
Figure 3
Figure 4

The obtained quality assessments coincide with the assessments of other modern wave model implementations (Li et

al., 2019, Reistad et al., 2011; Stopa et al., 2016). Good quality of the modeled data allows estimating the regular and extreme



characteristics of the wave climate, as well as the interannual variability of storm activity. We can conclude that
WAVEWATCHIII with set configuration adequetly represents real conditions of the wind wave fields of the Kara Sea.

**3. Results**

**3.1 Wave Climate**
The general features of the wave climate in the Kara Sea are discussed in this chapter.
The distribution of the maximum SWH and mean long-term SWH for the Kara Sea for the modeling period (1979–
2017) is shown in Fig. 5, a-d. The mean long-term SWHs are about 1.1–1.3 m (Fig. 5a) in the ice-free period. The maximum
mean SWH is 1.3 m and observed in the northern part of the Kara sea. This area is associated with the influence of storms
coming from the Barents Sea in the ice-free period. Formally the maximum SWH during the whole period reaches 9.9 m and
is observed in the northern part of the sea,  at the border with the Barents Sea (Fig. 5b). However, the wave conditions of this
area are largely determined by the Barents Sea and this area belong to the Kara Sea because of the formal border. In the central
part of the Kara Sea, the SWH maximum is 9.4 m and it is observed off the western coast of the Yamal Peninsula (Fig. 5b).
The maximum wave height of a 3% probability is 12.4 m (Fig. 5, c), and a 1% probability is 14.3 m (Fig. 5, d) for the central
part of the sea.
Figure 5
The Maritime Register Data (Wind and Wave ..., 2009) shows that the SWH with return period of 50 years is 5.4 m,
and for a 1% probability of exceedance it is 7.8 m. Our results differ strongly from these estimates. It is explained by the model
configurations and better wind forcing. Provided quality assessments for wave model results allow confessing the success of
this particular implementation. The modeling period is also important because the ice conditions become milder since 2009,
so the number of extreme storms increases (see next chapter).
A map of the long-term average probability of the ice occurrence is shown in Fig. 6, obtained from the
NCEP/CFSR/CFSv2 reanalysis data. This map is used for the analysis of the distribution of the maximum SWH and long-term
mean SWH. In general, the maximum values of SWH and mean SWH  are concentrated in the ice-free areas.
Figure 6
According to long-term mean SWH fields and to maximum SWH values, at least we can reveal two large regions
with particular spatio-temporal patterns of wave conditions, in the Kara Sea. The first one is the northern part which is often
occupied by the ice. It is affected by storms from the Barents Sea in the ice-free periods. The second region is the southwestern
part of the sea (Fig. 5-6). This region has longer ice-free period and wave generation occur without the influence of the Barents
Sea. It should be pointed that in the north-eastern part of the Kara Sea the influence of storms from the Barents Sea should be
expected, but, due to the high probability of the ice presence (> 0.8) in this region, the wave height is significantly lower than
in other parts of the sea.
The next step of our research was seasonal analysis of the SWH maximum for four periods: December-January-
February (DJF), March-April-May (MAM), June-July-August (JJA), September-October-November (SON). Figure 7 shows
the SWH maximum for different periods of the year for the entire simulated period. Seasonal maximal SWH variability is also
influenced by the ice conditions of the Kara Sea. Probability maps of the ice presence (with a concentration of more than 50%)
for the same seasonal periods according to reanalysis are shown in Fig. 8.
Figure 7
Figure 8
For the March-May period, the SWH does not exceed 4.5 m (Fig. 7a) due to the ice presence for almost the entire
period (Fig. 8a). The Kara Sea is free from ice in this period for a short time and in small areas. There is only one local SWH
maximum (8.1 m) in the southern part of the sea in June-August (Fig. 7b). During this period, wind speed is usually less than



in November-December therefore, severe storms are very rare despite the long ice-free period. Several SWH maxima are
observed in September-November, including the 9.4 m height in the central part of the sea (Fig. 7c). This maximum is an
absolute multi-year maximum for the central Kara Sea (Fig. 5b). Ice occurs only in the northern Kara Sea in this period (Fig.
8c). The strongest winds are observed in December -February. Most of the Kara Sea is ice-covered and the generation and
propagation of wind waves are limited. However, severe storms from the Barents Sea pass to the northern Kara Sea during
short ice-free periods. The absolute SWH maximum (9.9 m) for the entire sea was recorded was recorded there (Fig. 7a). The
differences in the wave characteristics in Figures 7a and 7d are mainly associated with the features of the atmospheric
circulation because the ice distribution is very similar in March-May and December-February  (Figs. 8a and 8d).
The mean and maximum values of the average wave period and average wavelength are presented below. The long-
term mean WP is 3.5 s (Fig. 9a). Such small WP is due to the long ice period and as consequence wave fetch is short. Mean
WP corresponds to the mean long-term SWH of 1 – 1.3 m. The maximum WP is 8.4 s for the central Kara Sea and 10 s for the
northern Kara Sea. the big WP is caused by several storms that come from the Barents Sea, where fetch is significantly greater
and swell has a longer period. The average wavelength is 30 m for the central Kara and 35–40 m for the northern Kara (Fig. 9
c). The maximum wavelength is from 160 m and up to 300 m (Fig. 9 d) for the central and northern Kara respectively. However,
large values are due to the calculating peculiarities of wavelength in spectral models. The remnant swell with an insignificant
wave height can provide a peak period of up to 20 s and a long wavelength in almost calm conditions, but these values should
not be considered as extreme.
Figure 9

**3.2 Storm recurrence**
The number of storm events per year was calculated in the Kara Sea according to the POT method (the technique is
described in Chapter 2.2). The events have different SWH thresholds from 3 to 7 m. Next, we will call these storm events with
a different wave height simply a storm. At first, we analyzed the number of storms for each year (Fig. 10), which called
recurrence of storm. Cases of storms with the SWH ≥ 3 m were observed about 30 times per year. Number of storms with the
SWH ≥ 4 m is about 15 times. The most severe storms with a threshold 7 m were not registered each year. The maximum
number of storms with SWH ≥ 3 m was in 2016. It is noteworthy, that in 2016 peaks were also registered for all other thresholds
and the recurrence of the most severe events ≥ 7 m is the highest. A local maximum number of storms with SWH thresholds
3 and 4 m was noted in 1995. The minimum numbers of storms for several SWH thresholds were noted in 1998 and 2003. A
linear positive trend in the number of storms is observed for almost all SWH thresholds. A double increase of  storm recurrence
was observed for cases with thresholds 3–5 m from 1979 to 2017. It's worth noting that there is high interannual variability in
the number of storms. The average variance is about 25–30 % from year to year.
The significance of trends was assessed by the F-test. Trends for the number of weaker storms more than 3–4 m are
significant at the level $p = 0.05$. For more severe storms with SWH thresholds 5–7 m, trends are statistically insignificant.
Similar result were obtained for 2005 to 2018 (Duan et al., 2019).
Figure 11
The analysis of the ice concentration variability in the Kara Sea was performed to explain the interannual variability
of the storminess. The graphs of ice probability for two points in the Kara Sea is presented in Figure 11. Ice probability is the
ratio of number of days with observed ice to the duration of the whole year. The points were selected in the central  and
southern parts of the Kara Sea to demonstrate the difference of  the ice conditions. There is a significant negative trend in the
variability of ice cover. Ice probability is approximately twice as less from 1979 to 2017. This trend is observed at both points.
It can be assumed that ice cover decreases in the whole sea. This fact has been detected by various researchers previously
(Cavalieri and Parkinson, 2012; Caian et al., 2018; Comiso et al., 2017; Maslanik et al., 2011; Serreze and Stroeve, 2015). The
ice probability decreases from 0.7 to 0.55 in center of Kara Sea (T1 point). The decrease is even greater in the southern Kara



(T2): from 0.7 to 0.4 (Fig. 11). A local minimum of the ice probability was noted in 1995 in the southern part of the Kara Sea.
This minimum probably led to an increase of the number of storms with SWH ≥ 3 and 4 m (Fig. 10). But in 1995 ice cover
reduction was observed only in the southern Kara, not in the whole sea, that's why such reduction does not cause extreme
storms (≥5 m). The maximum ice cover was observed in 1998–1999 and amounted to 0.8 in both points. It led to the storminess
weakening (Fig. 10). The ice probability minima were observed in 2012 and 2016 in T1 and T2. These minima coincide with
a significant increase in number of storms (including storms with SWH ≥7 m) that was observed exactly in these years.
Figure 11

Also, we analyzed the interannual variability of the wind conditions in the Kara Sea to explain the interannual

variability of storm recurrence. The relationship between wind speed and wave height is non-linear. In addition, we need to
consider such factors as fetch length, ice presence, and duration of wind impact. Therefore, correlation analysis for the wind
recurrence with defined speed (higher than threshold values) and wind duration time with the storm repeatability was
performed. The average daily wind speed at 10 m above the sea level was obtained from the reanalyses NCEP/CFSR and
NCEP/CFSv2 for the period 1979–2017 for two points (the same as for the ice probability analysis points): T1, 66.04 °E, 73.91
°N; and T2, 61.59 °E, 71.09 °N. The maximum correlation (0.65) is observed in a comparison of the number of storms with
SWH threshold 4 m and wind recurrence with speeds greater than 10 m/s, it was revealed for 2 continous days at T1. These
storm wind conditions were used as an indicator in the analysis of the interannual variability of the wind.

The recurrence of storm wind conditions, number of storms with SWH more than 4 m, and the ice probability are

shown in Fig. 12. The recurrence of storm wind conditions agrees quite well with the recurrence of storms. It is also seen that
years with high sea ice conditions reduce the number of storms in 1998 and 2003 despite the average values of storm wind
conditions. The significance of trends was estimated by F-test. Trends of the storm waves recurrence and the ice probability
are significant at the level of 0.05, and the trend of the storm wind recurrence is statistically insignificant.

Figure 12

Thus, there is a evident positive trend for number of storms in the Kara Sea according to the results of the analysis.

This trend is mainly caused by the sea ice cover decrease over the past 40 years, the trend of storm wind conditions is not
statistically significant. The interannual variability of events with SWH more than 3–4 m correlates quite well with the wind
recurrence (speeds more than 10 m/s). However, both wind and ice conditions, certainly affect the storminess. Ice cover
reduction leads to an increasing of weaker storms SWH ≥ 3–4 m in the southern Kara Sea. Such reduction in the entire sea
leads to the increase of extreme storms number (SWH > 5–7 m). The influence of ice cover variability also were obtained in
the work (Li et al., 2019).

Climate changes in storm wind conditions may be associated with changes in the ice conditions in the Kara Sea,

however, this analysis is already beyond the scope of our research and requires more detailed study. It is a challenge task for
future research.

**3.3 Probability analysis of storm waves in different sectors of the Kara Sea.**

Based on the analysis of the mean long-term and seasonal variability of the wave heights, the Kara Sea was divided

into several sectors with different wave conditions. In these areas, several zones of maximum waves are observed in different
periods of the year (Fig. 7a–d). This segmentation allows to analyze extreme storms in detail.

Figure 13

A catalog of storms with SWH more than 3 m was formed for each sector shown in Fig. 10. The POT method was

used to create the catalog, and a threshold of 3 m was chosen as the 95th percentile for the sample for the ice-free period. In
this catalog, each member of the series is a separate storm event. It is necessary condition for the independence of the members
of the series according to the method of "independent storms" (Cook, 1982). The length of the data series is sufficient for
statistical analysis. Series consists of 450–750 values depending on the sector.


The storm data series for each of the 6 sectors were approximated by various distribution functions. A comparison of the
functions with the empirical data showed that the best approximations for the storm recurrence was the Pareto distribution
$$F(H) = 1 - \left(\frac{H_{th}}{H}\right)^{\gamma},$$    (2)
where $H_{th}$ is the threshold value. $\gamma$ is the distribution parameter easily determined by the least square. For this purpose, formula
(2) by logarithm is reducing to
$$ln\big(1 - F(H)\big) = -\gamma \cdot \ln(H) + \gamma \cdot \ln(H_{th}).$$    (3)
If the empirical values on the diagram are located along a straight line in the logarithmic coordinates, this means that
the empirical distribution corresponds to the Pareto distribution. The quantitative correspondence of the empirical and the
theoretical distribution is established by using known statistical criteria.
Pareto distribution for all sectors is shown in Fig. 14. About 99% of the points are described by the Pareto distribution
with parameters $H_{th} = 3$ m and $\gamma = 4.8$ and a determination coefficient of $R^2 = 0.98$ in sector 6. This approximation is used as
base distribution. The Kolmogorov-Smirnov test also shows that the Pareto distribution is quite well. A similar pattern of
distribution functions is observed for all six sectors.
Figure 14
The average value of $\gamma$ is equal to 4.6 (varying from 4.2 to 5.0 for different sectors). The proximity of the parameters
in the Pareto distribution indicates that the extrema are generated in all sectors with a similar law. Thus, the wave generation
with an SWH more than 3 m is determined by the same mechanism. The basis of the hypothesis is the series of extrema
determined by the same law of probability distribution. A similar analysis is given in (Taleb, 2010). All extreme events are
called "swans", while the maximum and the largest rare events are "black swans". However, there are very rare cases when
the empirical distribution deviates and exceeds the base distribution in the large values area. These unique events are called
"dragons" (Sornette, 2009). The extreme values of SWH greater than 8 m observed in 2, 4, 5 sectors.
In our case, several extreme values that deviate from the base distribution were detected in each sector due to the
analysis of the distribution functions. The direction is common to these deviations - the points always deviate upward (Fig.
14). Therefore, these events are "dragons". A similar principle was used for freak wave detection (Buhler, 2007) and in
studying wind speed extremes (Kislov and Matveeva, 2016; Kislov and Platonov, 2019; Platonov and Kislov, 2019 ). Unique
extrema "dragons" falling out of the base distribution and have a different distribution law and, probably, a different genesis.
It is very important that the probability of extreme events based on a theoretical function, in our case, the Pareto
distribution. For example, the data of sector 6 (East coast of Novaya Zemlya) (Fig. 14) shows an SWH equal to 6.7 m (logarithm
1.9) – almost the last value that still lies on the base Pareto distribution. This value repeated through 47 sample elements
$\left(\left(\frac{H}{H_{th}}\right)^{\gamma}\right)$ on average. However, storms with such SWH occurs about a 100 times in reality (Fig. 14), twice as much as it was
planned by the Pareto approximation. A similar situation is reflected for the other sectors in the "dragons" zone. Use of base
distribution in this zone leads to incorrect probability calculation results. This fundamental result demonstrates the source of
systematic errors in evaluating the recurrence of extreme wave heights, which are especially relevant in applied and forecast
tasks.
The probability of "dragons" doesn't match the base distribution. In the Kara Sea, the occurrence of storms with high
waves depends on several factors simultaneously: primarily on the wind speed, direction, and duration of the wind, secondly
on the ice conditions (fetch limit) or the influence of the Barents Sea (for 4-th sector). The  number of storms with SWH more
than 3–4 m is closely related to wind speed and wind duration as it was shown in chapter 3.2, but the repeatability of storms
with SWH more than 6–8 m requires the simultaneous combination of small ice cover and extreme wind conditions. Thus, the
division of the empirical distribution function between "black swans" and "dragons" occurs when the influence of small ice
cover (and consequently more long fetch) is observed besides the wind forcing. Since wind and ice conditions are considered
as approximately independent events, their joint probability is much lower than the probability of rare wind events.





Extreme events with any (even very large) wave height can occur according to the base distribution function, formally.
However, the empirical function for the "dragons" is nonlinear and goes to a certain plateau; it was shown in the logarithmic
graphs. The $\gamma$ values (starting with some values of $H$) begin to increase rapidly. Thus, there is a certain natural limit observed
for extreme events. "Dragons" have a limitation for maxima wave height that differs from the base distribution. The basic
distribution ends in the range of SWH values 6.5–8 m in different sectors. Such differences are associated with the definition
of freak waves in the article (Buhler, 2007). Freak waves are unique anomalous individual waves that do not correspond to the
general distribution. In our case, we have a similar picture on the synoptic scale, where specific storms with a certain SWH
maximum defined as "dragons".
Figure 15 shows the graph of "dragons" passing by year in each of the six sectors. This graph was analyzed for the
possible impact of climate change on the "dragon" recurrence. "Dragons" occurred in sectors 1 and 5 only after 1997-2000,
when the increased recurrence was registered for the entire Kara Sea. A higher recurrence of "dragons" was registered in years
when there were simultaneous peaks of wind recurrence and small sea ice cover (see Fig. 13).
Figure 15
Thus, a reduction of the sea ice cover and increased recurrence of stronger winds lead to an increase of the extreme
wave heights. There are climatic changes in the "dragon" repeatability, that increased due to the ice cover reduction in the last
40 years.

**Discussions and Conclusions**

A wave climate and storm recurrence in the Kara Sea has been presented based on the results of wave modeling. The
SWH, the mean WP, and mean WL fields were obtained for every three hours from 1979 to 2017 (39 years in total). The mean
SWH for the entire sea varies from 1.1 to 1.3 m. The SWH maximum is 9.9 m and it is observed in the northern part of the
Kara Sea. Analysis of maxima for different times of year showed that the SWH does not exceed 4.5 m in March-May. The
wave generation is limited by the ice presence in some periods of the year. The long-term mean wave period value is 3.5 sec
and the average wave lenght is 30 m for the central Kara and 35–40 m for the northern part.
The storm recurrence with an SWH threshold from 3 to 7 m was calculated in the Kara Sea for each year according
to the POT method. Storms with an SWH ≥3 m are observed about 30 times per year on average, with SWH more than 4 m -
about 15 times. Storms with a threshold of 7 m are not observed every year. The storminess was higher in 1994–1995 and after
2008. The minimum numbers of storms were registered in 1998 and 2003.
The combined analysis of the storm activity, the recurrence of strong winds, and the ice probability was carried out.
The high recurrence of strong winds and the absence of sea ice lead to increase of storm number with SWH 3–4 m in the
southern Kara Sea. When the sea ice probability decreases for the whole sea and recurrence of strong winds is high
simultaneously, then the number of extreme storms (SWH more than 5–7 m) increases.
There is an obvious positive trend of the storm activity in the Kara Sea and a positive linear trend of the weaker storm
recurrence (SWH more than 3–4 m) for 1979 – 2017. Linear trend of the severe storm recurrence (SWH more than 5–7 m) is
positive but statistically insignificant because such events are rare. This trend is mainly caused by a reduced sea ice cover over
the past 40 years, the trend in recurrence of storm wind conditions is not significant.
The Kara Sea was divided for six sectors with different wave conditions due to the analysis of the mean long-term
and seasonal variability of wave heights.
The probability analysis for the six sectors of the Kara Sea was provided. Different approximations were compared
with the empirical distribution, the best approximation for the storm recurrence was the Pareto distribution. The proximity of
the parameters in the Pareto distribution indicates that the extrema generation occurs in the same way for all sectors.


Analysis of the distribution functions for each of the sectors showed that several extreme events ("dragons") deviate
upward from the base Pareto distribution. Thus, the division of the empirical distribution function between "black swans" and
"dragons" occurs when the influence of small ice cover (and consequently more long fetch) is observed besides the wind
forcing. "Dragons" occurred in sectors 1 and 5 only after 2000, when the increased recurrence was registered for the entire
Kara Sea. A higher recurrence of "dragons" was registered in years when there were simultaneous peaks of wind recurrence
and small sea ice cover. On a time scale of 40 years, we see climatic changes in increasing the recurrence of such extreme
events as "dragons".

There are some questions of the quality assessment of the wave model for extremely high waves, but unfortunately,
we do not have full-scale direct measurement data in the Kara Sea, and satellite data also need verification and are not accurate.

In this paper, we do not consider the relationship between number of storms variability with global climatic indices
of large scale atmposphere circulation. Earlier in (Myslenkov et al., 2019), we showed that for the Barents Sea the number of
storms only of extreme events with SWH $\geq$ 7 m and only for the DJF has a low correlation with the Arctic Oscillations index.
And this is largely due to the decisive influence of the Atlantic on the Barents Sea. In the Kara Sea, the influence of the Atlantic
and Western transport is even less, therefore, probably there is no connection with global indices here. On the other hand, we
showed that the wave climate in the Kara Sea is regulated by ice cover variability. Connection between sea ice loss and the
Arctic Oscillation detected in (Yang et al., 2016), therefore theoretically it is possible to find a connection between wave
climate of the Kara Sea and global indexes, what are we going to do in the future.

**Data availability**
Data and results in this article resulting from numerical simulations are available upon request from the corresponding
author.
**Author contributions**
The concept of the study was jointly developed by SM. SM did the numerical simulations, analysis, visualization and
manuscript writing. VP and AK did the probability analysis of storm waves and its visualization. KS analysed the results of
numerical modeling. IM did the validation of the model. SM prepared the paper with contributions from VP, AK, KS and IM.

**Competing interests.**
The authors declare that they have no conflict of interest.


**Acknowledgments.**
The wave modeling and probabilistic analysis were done with the financial support of the RFBR (project 18-05-60147
Myslenkov S.A., Platonov V.S., Kislov A.V.).
Data analysis funded by the Ministry of Science and Higher Education of Russia, theme 0149-2019-0004, and RFBR project
20-35-70039 (Silvestrova K.P).
Validation of the model done by I.P. Medvedev with the financial support of the RFBR (project 18-05-60250).
Authors gratefully thank A.Yu. Medvedeva for the editorial remarks.

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

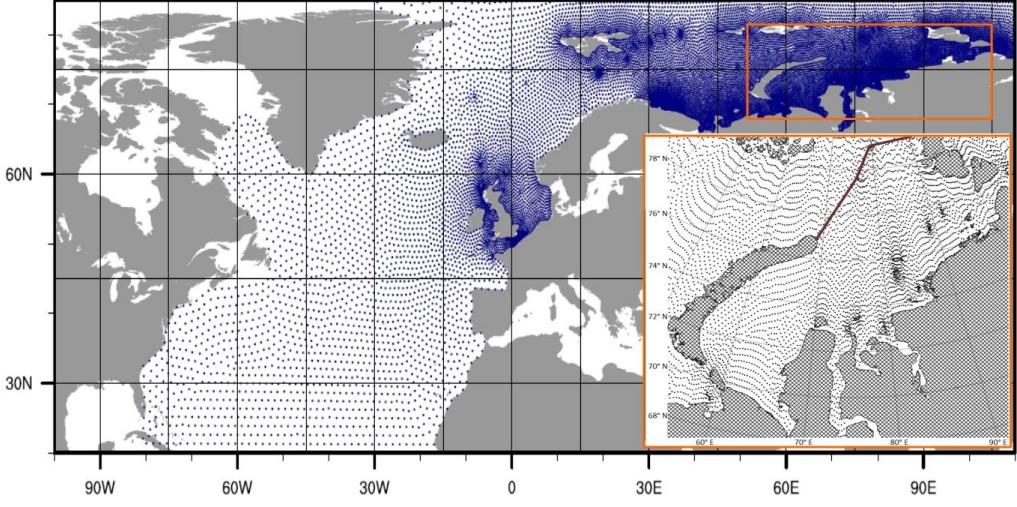


Figure 1. Unstructured computational grid for the North Atlantic and the Kara Sea.

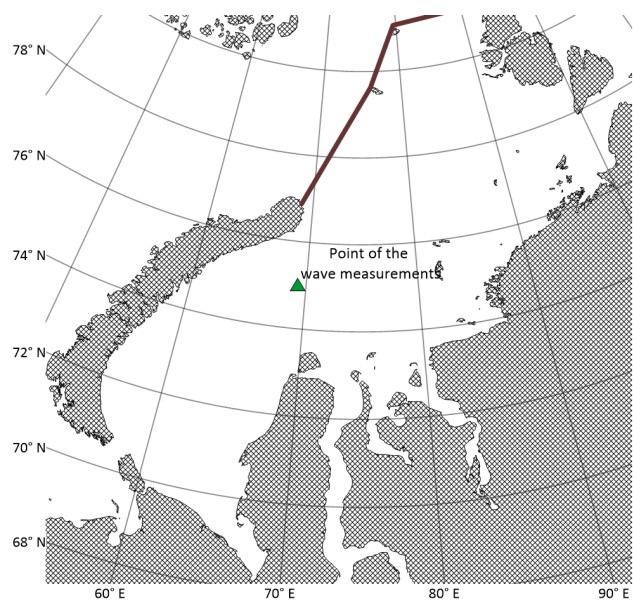


Figure 2. Location of the wave measurement stations (a), a histogram of the distribution of wave heights at station No. 3 (b).

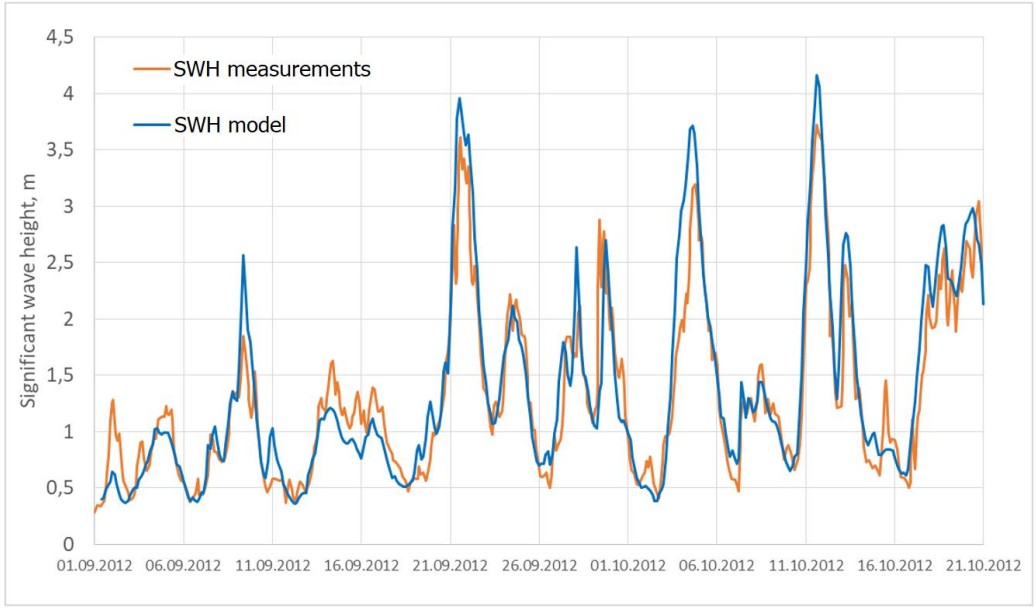


Figure 3. The measured and simulated SWH for station No. 5.


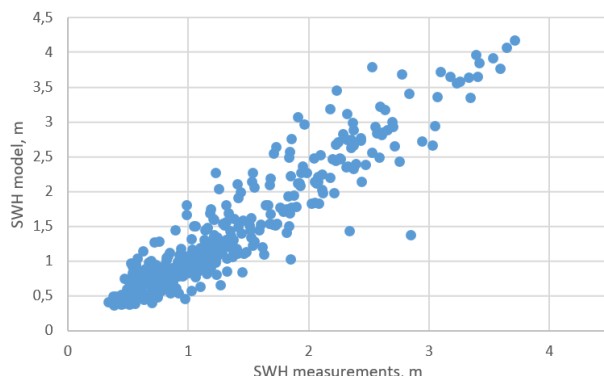


Figure 4. Scatter diagram of measured and simulated SWH for all points



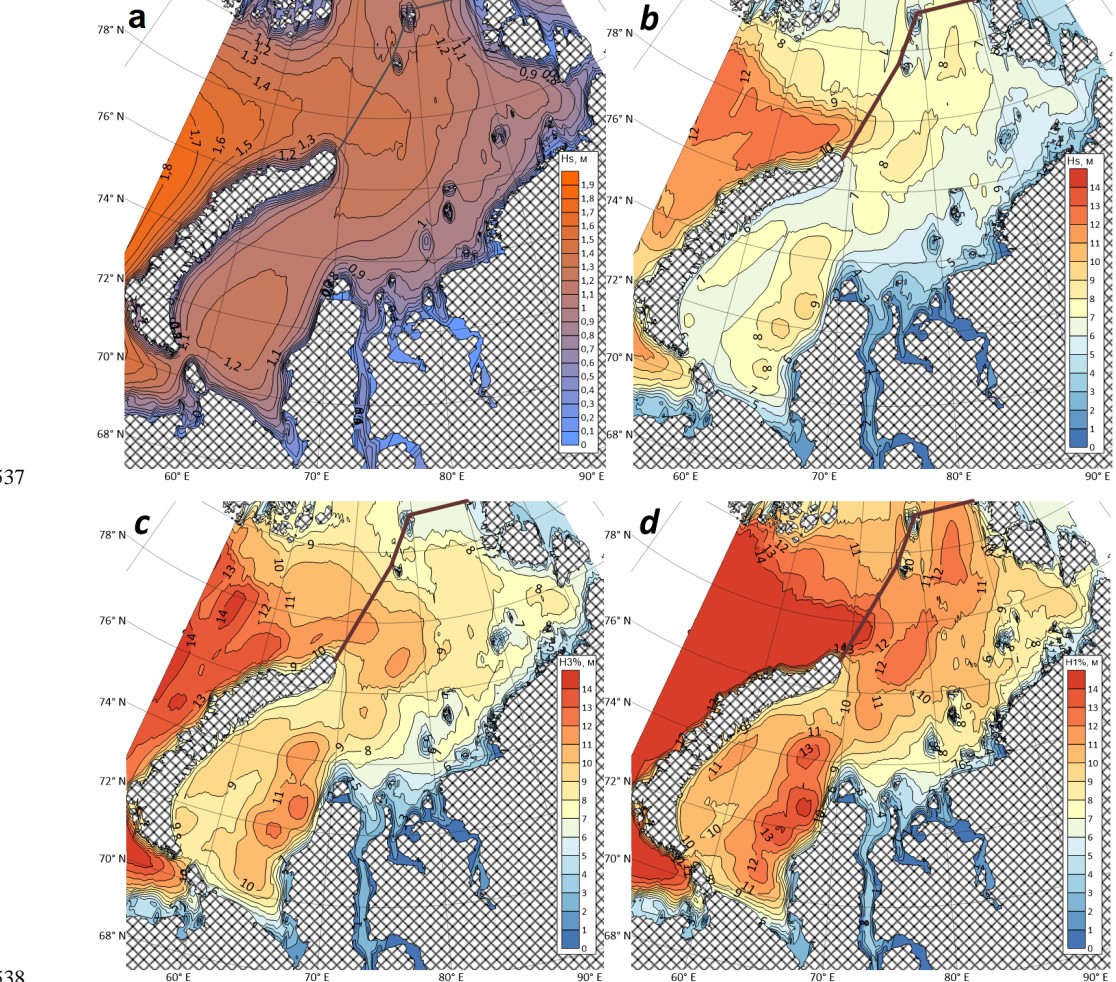



Figure 5. The long-term mean (a), maximum (b) significant wave heights, maximum wave height of 3% probability of
exceedance (c), and maximum wave height of 1% probability of exceedance (d) according to the modeled data in the Kara Sea
for the 1979–2017 period.

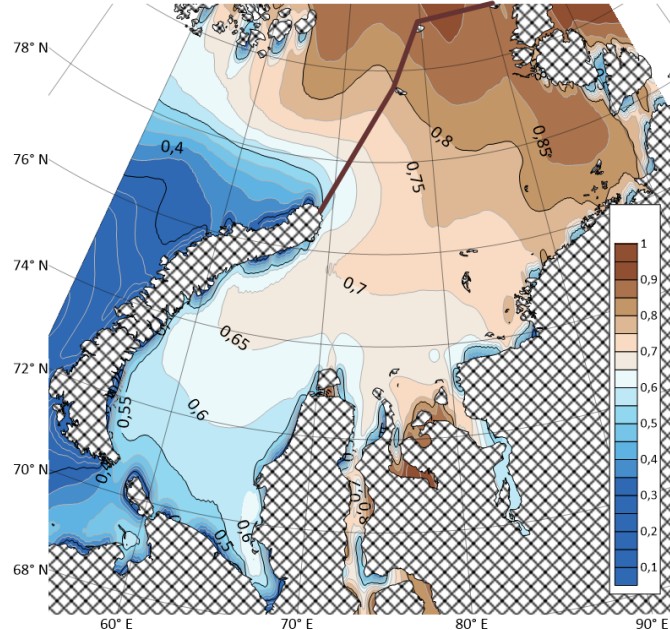


Figure 6. The long-term average probability of the ice presence of with a concentration more than 50% in the Kara Sea

according to reanalysis data from 1979 to 2017 (in 0–1 unit).



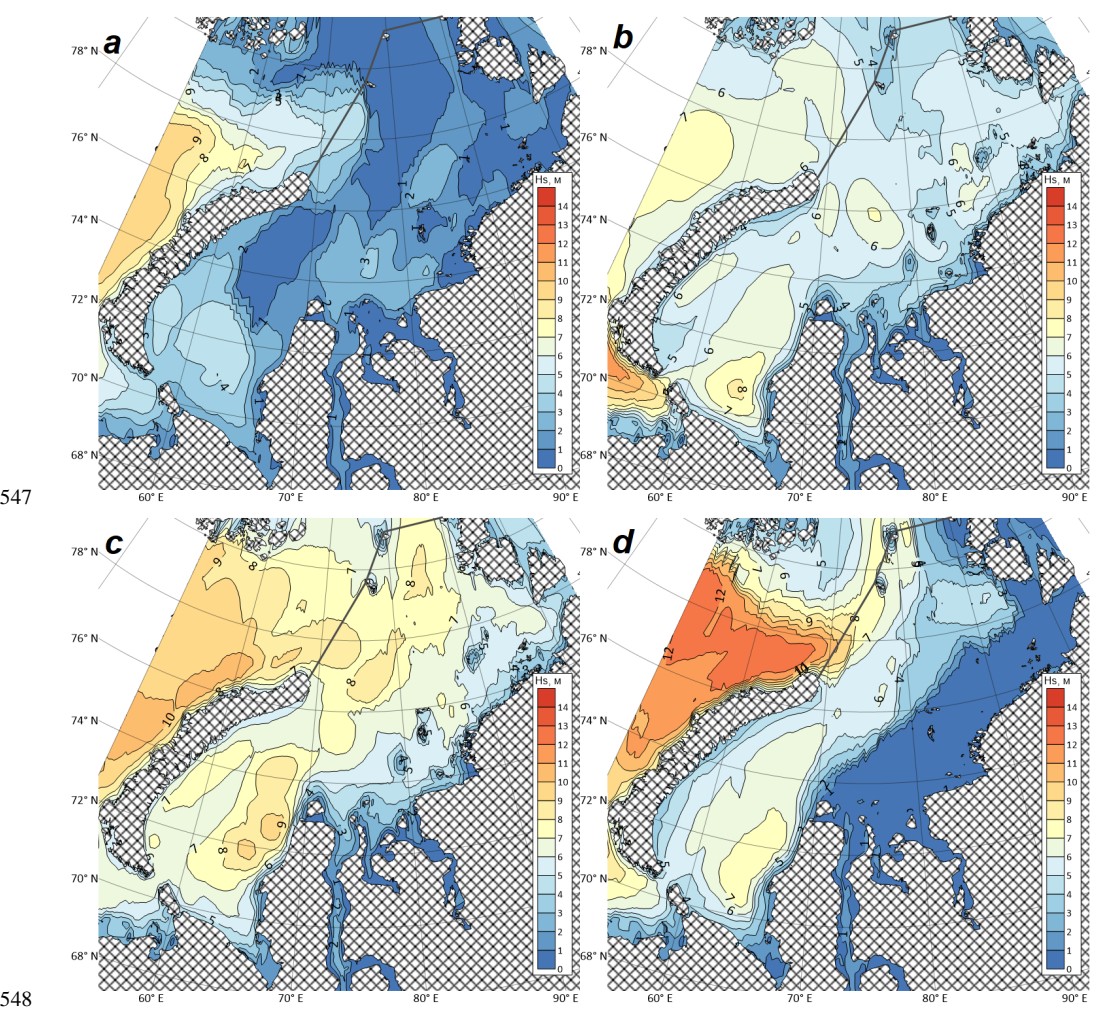


Figure 7. The maximum SWH in the Kara Sea according to the model data (from 1979 to 2017) for the periods: MAM (a),

JJA (b), SON (c), DJF (d).



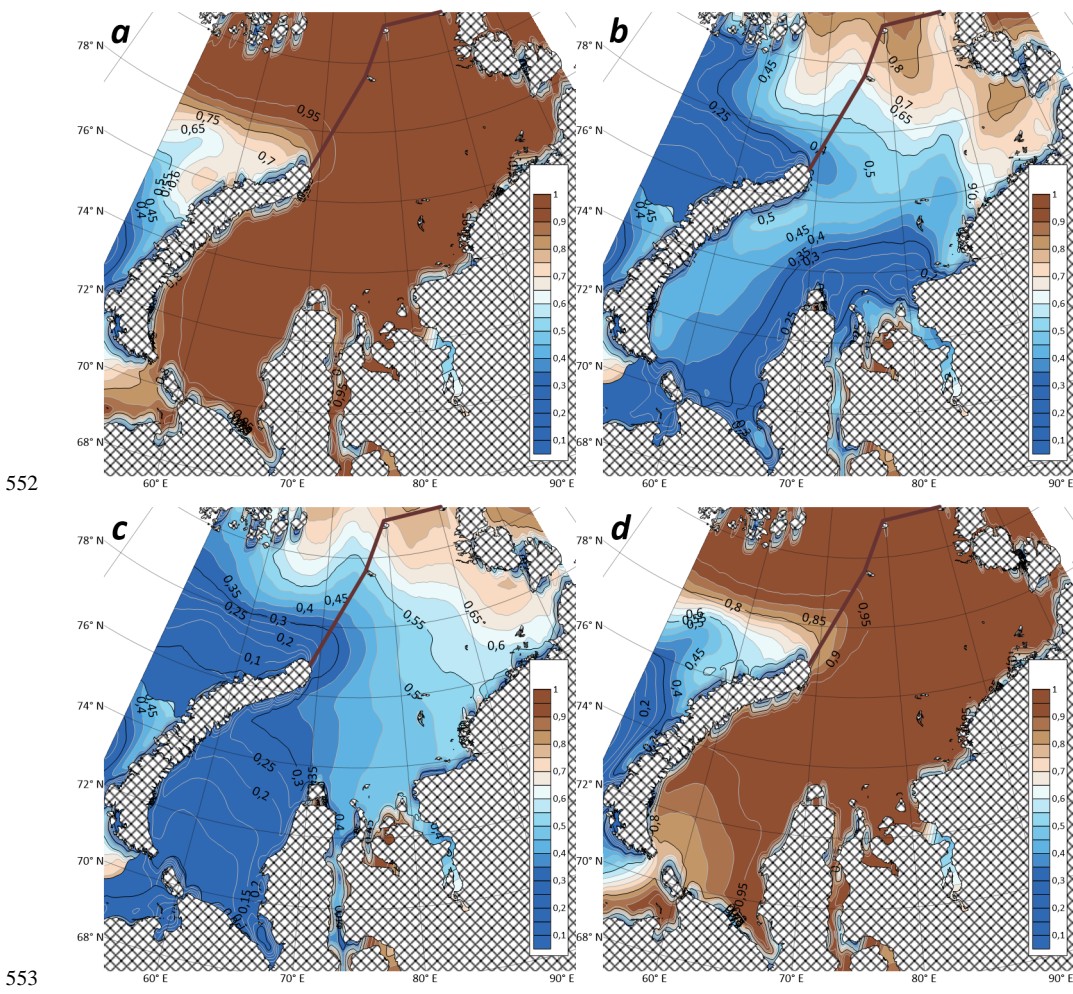



Figure 8. The probability of the presence of ice with a concentration of more than 50% in the Kara Sea according to reanalysis
data (in 0–1 unit) for the periods: MAM (a), JJA (b), SON (c), DJF (d ).



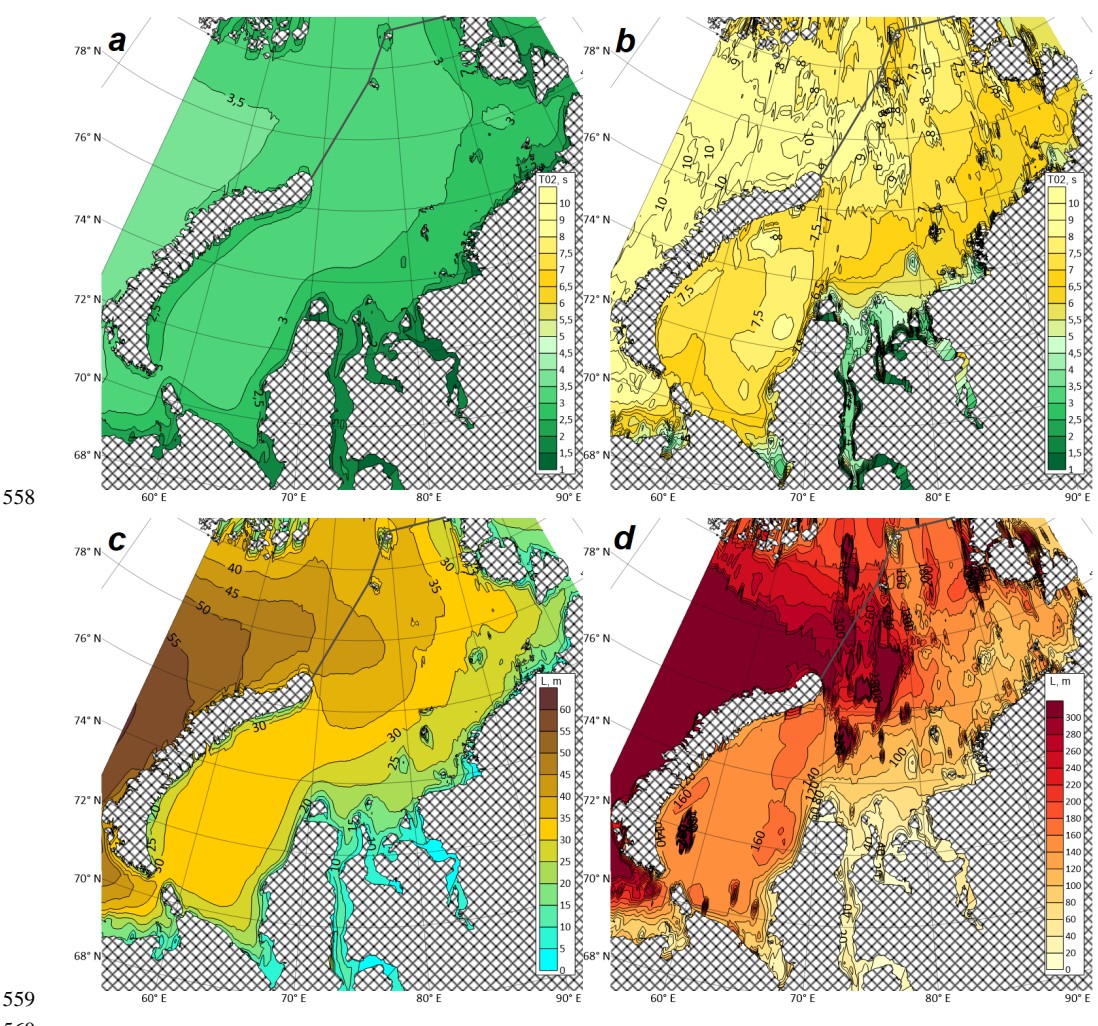

Figure 9. The long-term mean period (a), the maximum period (b), the long-term mean wavelength (c), and the maximum
wavelength (d) in the Kara Sea according to modeling data for the period from 1979 to 2017.


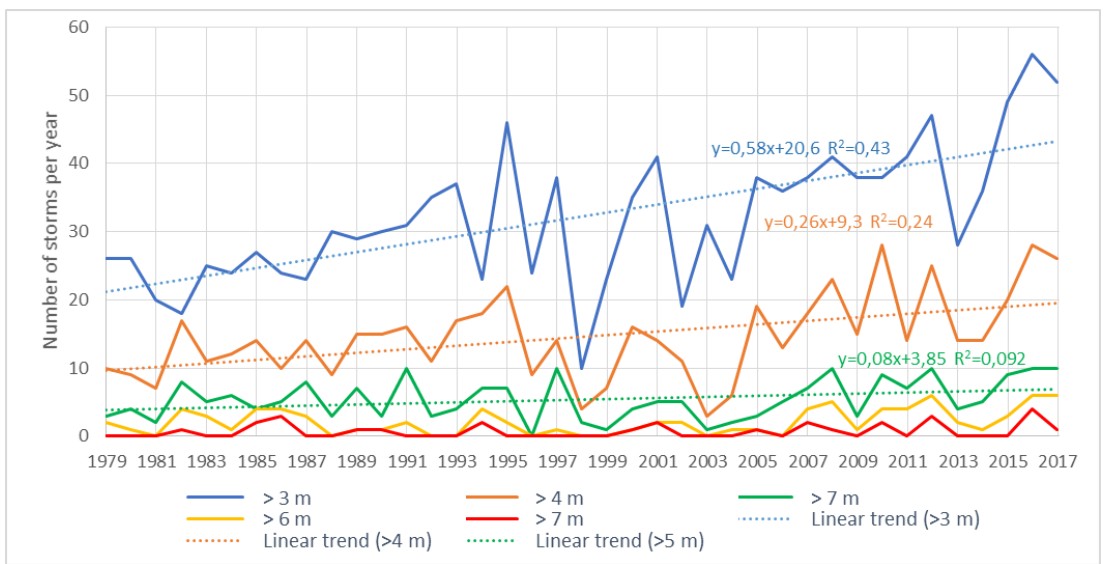



Figure 10. The number of storms with different thresholds per year and its linear trends for 1979 to 2017.


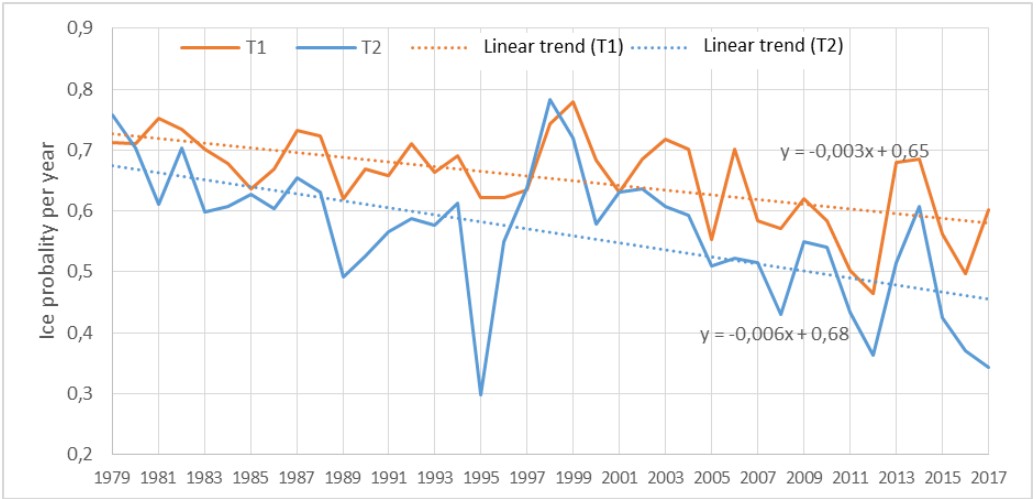


Figure 11. The probability of the ice presence with a concentration of more than 50% for two points in the Kara Sea by years.
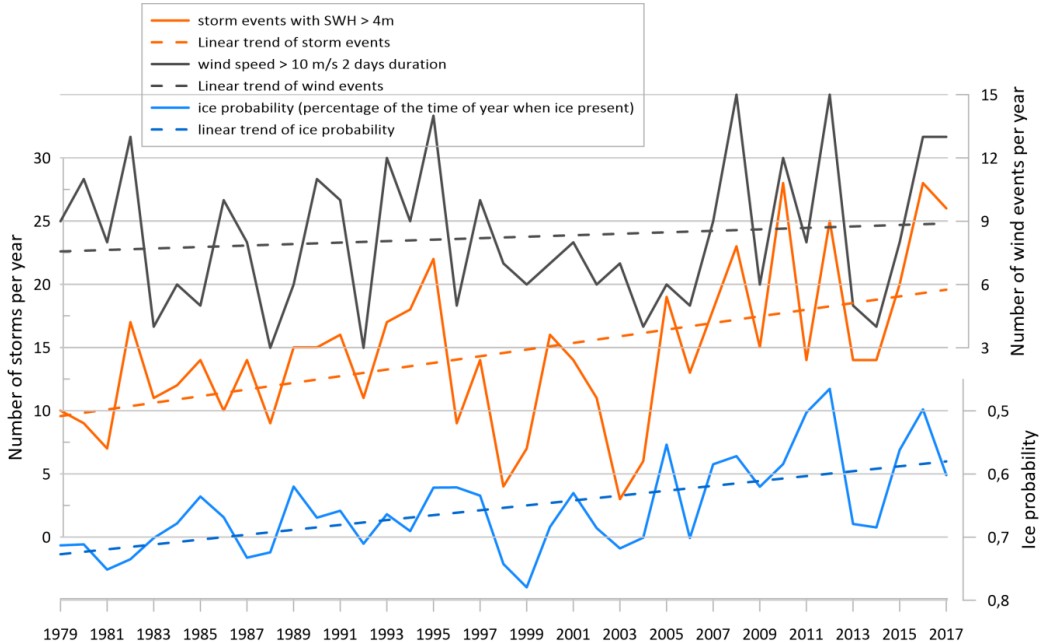


Figure 12. Recurrence of wind speed of more than 10 m/s and 2 consecutive days at point T1, the number of storms with a

threshold 4 m, and probability of the ice presence in point T1(opposite scale).


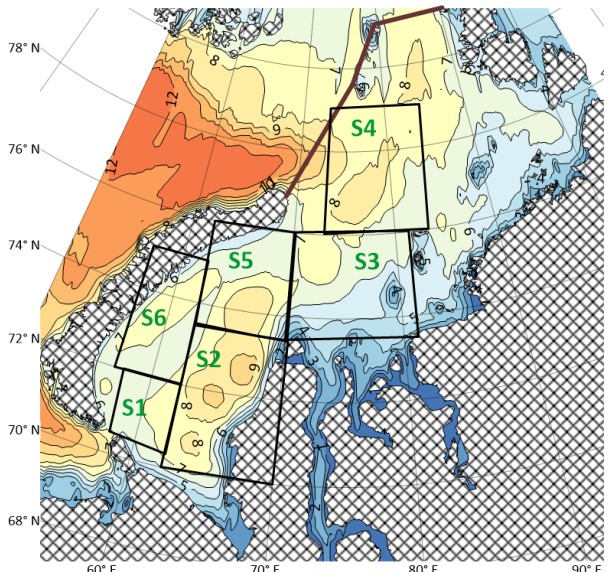


Figure 13. The SWH maximum and segmentation of the Kara sea: six sectors with different wave conditions.



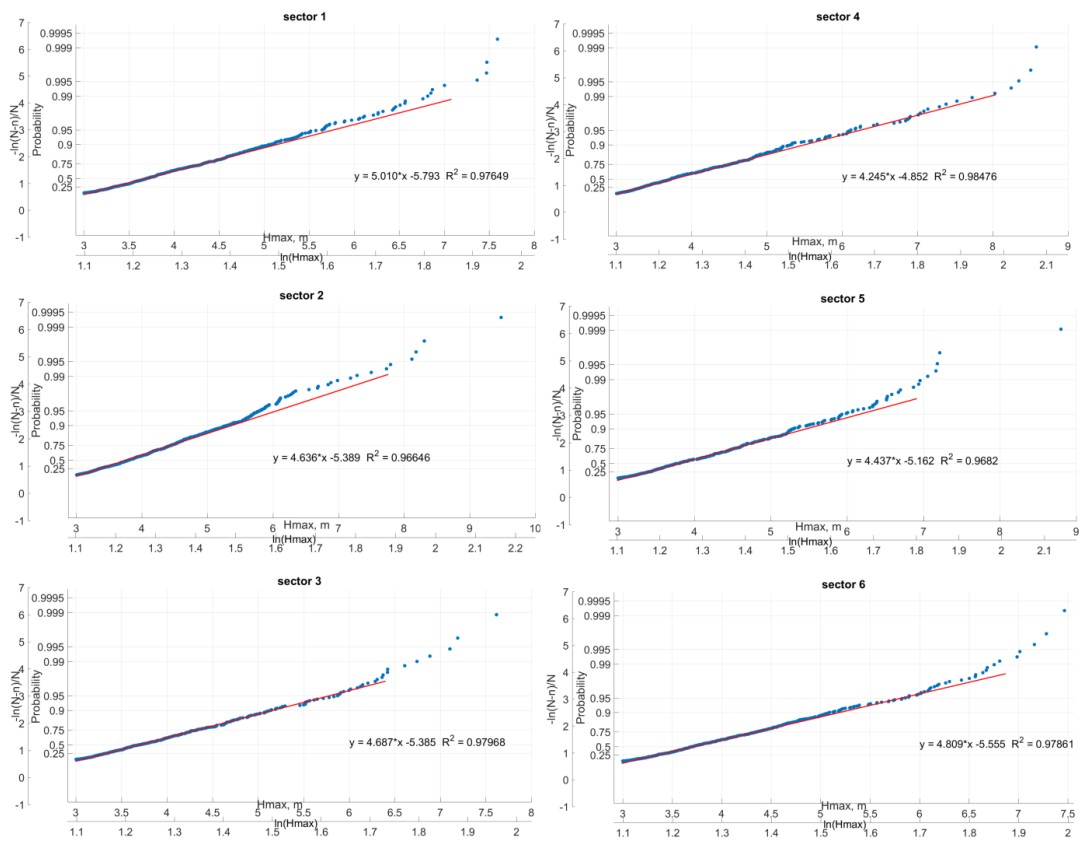

Figure 14. The empirical probability distribution of storms with different wave heights for each of the six sectors, presented in the Pareto logarithmic coordinates. The coefficient of determination and regression equations are given for each sector.





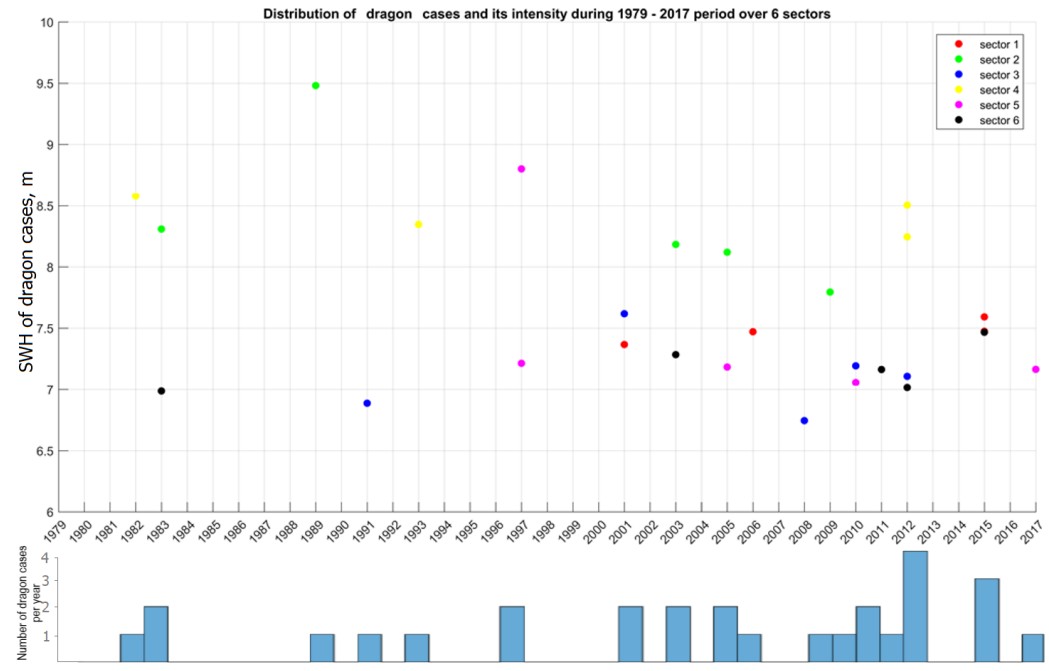

581

582 Figure 15. Cases of extreme events (dragons) from 1979 to 2017 for all sectors of the Kara Sea.

583