# Peer review of "Wave climate and storm activity in the Kara sea"

_Natural Hazards and Earth System Sciences, 2020_

## Referee Comment (RC1) · Anonymous Referee #1 · 3 Aug 2020

This manuscript describes the wave climate in the Kara sea as obtained from a 39-year long simulation using the wave model WAVEWATCH III. After introducing their motivations, the authors describe the configuration used to run their hindcast. They also give the details of the method used perform the storm activity analysis as well as a quick assessment of the wave model results. Using their simulation outputs, they perform a statistical analysis on the sea state over the 1979-2017 period with a particular focus on the wave climate and the occurrence of extreme events (high waves due to storms). They discuss the interannual trend and variability of the occurrence of storms, each storm event being categorized depending on the maximum wave height it is associated with. They find an increase of storm events in the Kara Sea over the studied period and show that the increase in the number of storms generating waves higher than 4m is correlated with the decrease in the ice cover. They then divide the studied domain into 6 sectors and show that the recurrence of storms obeys a Pareto distribution for most events. The distribution of events associated with the highest wave height does not however fit this Pareto law, and the authors classify them as "dragons". The probability of "dragons" is higher than foreseen by the Pareto distribution, and seems to be related to the decrease in the ice cover.

The statistics of extreme events is not my domain of expertise. I am however used to performing statistical analysis on wave model outputs, particularly in the Arctic, and my comments are below:

Although some of the results described by the authors could be worthy of being published, I cannot recommend this manuscript for publication in its current state for the following reasons:

- The choice of the parameterization should be more justified. Why are the authors not using a more recent version of WW3? It would offer different ways of accounting for the presence of sea ice, and thus could provide an estimate of the sensitivity of the results to the representation of wave-ice interactions. The manuscript also lacks a discussion on the sensitivity of the model to the choice of the parameterization.

- The validation of the model is very "light", while the quality of the sea state reproduced by the model is absolutely key to support the results described further in the manuscript.

- The manuscript is very hard to read and follow. The language used is very vague, and the English is often very confusing. For instance, the authors keep referring to "thresholds", "distributions", but often forget to mention "of what", forcing the reader to guess what the authors are referring to. The structure of the text should also be revised to emphasize more explicitly the authors' motivations and key results. The number of typos (missing verbs, misplaced brackets…) is also very high, which contributes to giving the impression of lack of rigour. I recommend a thorough rewriting of the entire manuscript before any resubmission.

I will try to be a bit more specific section per section:

**Abstract:**
Overall, the level of detail in the description of the results is very high, while the motivations for the study and the importance of the results remain unclear.

P1L16: *with different thresholds (from 3 to 7 m)*
Thresholds of what?

P1L2:3 *If in the entire sea the ice cover decreases that leads already to increase of the extreme storms.*
This sentence is very hard to follow, please use commas and check the English.

**Introduction:**
I get that interactions between waves and sea ice are not the core of your paper, but I would emphasize a bit more the challenge that sea ice represents for wave modelling. The quick change in sea ice conditions is what makes this study interesting.

P1L37->P2L44: I feel like the same arguments are repeated in every sentence, once should be enough.
P2L50 *data from models are preferable.*
They are also limited by the presence of sea ice, as waves-in-ice modelling remains quite challenging (e.g Squire, 2020). Also, I would not say that model data are "preferable" to observations.

P2L56 Please mention the method followed by Stopa et al.
P2L78 *These highlights [...]*
I don't understand this sentence.

**Data and methods:**

P3L98 Why do the authors use WW3 v4.18 from 2014 and not a more recent version (5.16 or 6.07)? Version 4.18 is very limited in its ability to represent waves in ice

P3L110: As mentioned earlier, I would like more justifications for the choice of the parameterizations. Why ST1? It uses the same scheme as the 1$^{st}$ version of the WAM model at the end of the 80's. There have been some improvements since.

P3L110: IC0 simply considers an ice-covered grid point as land, this is the simplest solution of all. I also do not understand the comment on the exponential attenuation.

P4L132 In a spectral wave model, SWH is not computed as the average height value of the 33% highest waves.

P4L142 *within the sea sector*
What is the sea sector?

P4L144 *with waves more than this threshold.*
Please rephrase.

P4L150->153 All this paragraph is very confusing. It should start with the motivation (deep analysis of the extreme event with the SWH >3m), and the details should be rephrased to be less ambiguous (threshold of? What is the central part of the sea?).

P4L156 There is no conjugated verb in this sentence.

Fig2b is missing on my version of the manuscript.

P5L169 *We can conclude that WAVEWATCH III with set configuration adequately represents real conditions of the wind wave fields of the Kara Sea.*

I strongly disagree with this statement. Overall, I find section 2.3 very weak. Even with the panel (b) of Fig. 2, the evaluation of the simulation on which all the analysis relies is performed with only one mooring, over one month? This is far from being sufficient to me.
Moreover, the chosen period for the evaluation (September-October) corresponds to the ice-free period in the Kara Sea, which is likely to be the one for which the model performs the best. This quality assessment also ignores the effect of the forcing of the ice conditions: how do they affect the quality of the model? In the results, there is a lot of discussion about the links between wave height and ice conditions. In these conditions, the quality of WW3 outputs in winter should be evaluated as well.

**Results**

The analysis performed is interesting, and the figures are nice and readable. However, as written previously, I strongly disagree with the comments stating that the quality assessment is successful (P5L188). This section is also very hard to follow due to the presence of confusing expressions (for instance: "ice conditions become milder" P5L189; "allow confessing (?) the success" P5L188, "high sea ice conditions" P7L273 …). I would also recommend avoiding the use of words like "obvious", "easily" and statements like "it can be assumed that ice cover decrease in the whole sea" (P6L252).

P8L318 and P8L324: These two sentences lack a conjugated verb.

**Discussions and Conclusion**

This section is very long given that it is mostly a summary of the Results and the discussion part is very short. I agree that the quality assessment of the wave model for high waves is difficult. I also agree that wave height estimated from satellites might be inaccurate and not available for all seasons and the whole period. However, these limitations cannot justify a 39-year long simulation being considered as validated with a quality assessment over 2 points in September-October 2012 only.

References:

Ocean Wave Interactions with Sea Ice: A Reappraisal
Vernon A. Squire
Annual Review of Fluid Mechanics 2020 52:1, 37-60

---

## Referee Comment (RC2) · Anonymous Referee #2 · 10 Aug 2020

Review comments for "Wave climate and storm activity in the Kara sea" by Stanislav Myslenkov, Vladimir Platonov, Alexander Kislov, Ksenia Silvestrova, Igor Medvedev submitted to Natural Hazards and Earth System Sciences (nhess-2020-198)

The manuscript presents a long-term modeling data set for the Kara Sea that has been generated on an unstructured grid. The data set is used to study the mean and extreme values of the wave parameters, their statistical change in time, and their relation to the wind regime and changing ice conditions. The authors find that the peak-over-threshold data of the storms follow a Pareto distribution closely for the milder events, but the highest events deviate strongly from this theoretical distribution.

The content of the paper is interesting and suitable for the journal. Nonetheless, the methodology, and the treatment and presentation of the results are somewhat lacking.

[Figure]

I therefore have to recommend that major revisions are made to the manuscript before it's suitable for publication.

Major Comment #1:

You use an old physics package and have very little measurement data to validate the model. The ST1 in essentially the physics that was used in WAM cycle 3, and it's from the 1980s. WAM cycle 4 physics were introduced already in the 1990s. Typically WW3 users use either ST4 or ST6, which are more modern and developed (as I understand it) specifically for WW3 (although they have seen implemented in other models also). I'm not an expert on WW3, but you seem to use an old version of the model code. It's a real shame that you have gone through all this work to produce a long data set with an unstructured grid, but used outdated physics.

I'm not quite sure what to do here, since rerunning everything with an more appropriate model setup seems somewhat unreasonable. I guess that, ultimately, this is not a fatal flaw, but the shortcomings and reasons for the choices should be openly and thoroughly presented. The ST1 dissipation for example dissipates swell through the whitecapping formulation in mixes sea-swell conditions. This is probable relevant for your case. The possible shortcomings in the model setup brings us to the next point.

The validation of the model is very light. I can understand that good observations are hard to come by, but is there no other measurements available? No remote sensing data? Please provide some references that the satellite data are "not accurate" (L390). Satellite data are routinely used, so is there some special circumstances here? The validation you have shows that the model overestimates the highest values, but this is completely brushed over. The entire model validation needs to be reworked here.

Major Comment #2:

With POT data one usually start with the Generalized Pareto Distribution (GPD), which has a variable shape parameter (see e.g. Coles, 2001 or the wide body of original

articles available). This is the standard methodology, and while some other distribution can definitely be chosen (by for example fixing the shape parameter in the GPD), this needs to be explored since it could better account for the highest values that doesn't seem to follow the distribution you chose.

Major Comment #3:

The references and general treatment of the topic is not as rigorous as one could hope, and the language also need to be modified to make aspects clearer. Grammatical issues can be fixed in a language check. One of my biggest objection is that you use the reference to Taleb (2010) with respect to your extreme value analysis. This is a popular book. Please cite actual scientific literature. Also in the wave modelling part there are several citations missing and it is just brushed over as being "standard". Also the reference to the "F-test" leaves the reader a bit unsure of what actual test is being used etc.

What is lacking more than anything is a critical view of the approach and how the shortcomings might influence the results, and a stronger connection between your results (and methods) and other existing scientific studies.

Minor Comment #1: L11, I believe the correct name is "WAVEWATCH III", not "Wave-WatchIII"

Minor Comment #2: L17, I don't understand what a "double growth" is?

Minor Comment #3: L27, To me it's not clear what it means that "99 % of the points are described by a distribution".

Minor Comment #4: L29, "twice as less" is not proper English. Please rephrase.

Minor Comment #5: The introduction is very "choppy" with short paragraphs listing different studies. A more proper way would be to summarize the themes in those studies in a way that put the current paper in perspective and support those points with references. The difference is obviously not clear cut, but in this case the readability

suffers.

Minor Comment #6: L100-101, WAVEWATCH solves the action balance equation, not the energy balance equation. Also, you are missing a dot product in Equation 1.

Minor Comment #7: L110, mention the references for the ST1 physics package (it is essentially WAM cycle 3 physics). This will make is readable also for modellers not familiar with WW3 specifically.

Minor Comment #8: 111-112. The correct reference for DIA is Hasselmann et al. (1985).

Minor Comment #9: L118 "Standard JONSWAP scheme". Please provide a reference even though it is "standard".

Minor Comment #10: L131, As a result of what? The wind information was every hour, so you I guess this is just as a result of you choosing to output every three hours (which is totally fine)?

Minor Comment #11: L132, Significant wave height in models is not H1/3.

Minor Comment #12: L 133-134. Does the 1% probability of exeedance mean that 1% of the single waves are higher than this threshold during the 3 hour period, or that the threshold for a single wave is exceeded, on average, once in every 100 three hour block. The language is a bit ambiguous.

Minor Comment #13: I don't see a panel b) with a histogram in Figure 2. Also, can't the location of the wave buoy just be marked in Figure 1? You are referring to stations No. 3 and No.5, but only show one station. The scatter plot also says "from all points". Do out have data from several stations?

Minor Comment #14: L169-170, It looks like the highest wave heights are being over-estimated, and this should be addressed since you are explicitly studying high events. Add a 1-1 line on the scatter plot to guide the eye.

Minor Comment #15: L186, The 50 year return period where? Entire Kara Sea?

Minor Comment #16: L225-228, So are you calculating the mean period (or actually the spectral zero-crossing period), but then using the peak period to calculate the wavelength. Why use different parameters? And the calculation of the wavelength should be made clear in the methods section.

Minor Comment #17: L243, Is there only one "F-test"? I have a feeling that this term encompasses many different specific tests. Please elaborate.

Minor Comment #18: L248, "for two points". Please mark the points on some map.

Minor Comment #19: L264. You say that you need to consider fetch length, but then you don't incorporate it into your analysis. I'm not saying you have to include it, but then word the sentence a bit differently, since now the reader is expecting it to show up.

Minor Comment #20: L269, Why is a wind speed exceeding a threshold for two whole days relevant when you have a minimum imposed distance of only 9 hours between storm events?

Minor Comment #21: L307, "About 99% of the points are described by the Pareto distribution". What does this mean?

Minor Comment #22: L332, "The probability of "dragons" doesn't match the base distribution." This might be because your base distribution is not suitable. You need to motivate your choice here and show how other distributions fit the data.

Minor Comment #23: L343-345, "The basic distribution ends in the range of SWH values 6.5–8 m in different sectors. Such differences are associated with the definition of freak waves in the article (Buhler, 2007)." I don't understand this. "Freak waves" (usually) refer to individual waves that are unusually high with respect to the underlying significant wave height. It has nothing to do with high significant wave heights. The phrase "are associated with the definition" is very vague. The author is Bühler, not

Buhler.

Minor Comment #24: Discussion and Conclusions. This is more of a long summary. What I would have liked to see would have been a critical discussion about the restrictions and the applicability of the results, an a better comparison to other similar results in the scientific literature.

References:

Hasselmann et al. (1985): "Computations and Parameterizations of the Nonlinear Energy Transfer in a Gravity-Wave Specturm. Part II: Parameterizations of the Nonlinear Energy Transfer for Application in Wave Models", JPO, Vol 15. Issue 11. pp. 1378-1391.

Coles, S.: An Introduction to Statistical Modeling of Extreme Values, Springer, London, https://doi.org/10.1007/978-1-4471-3675-0, 2001.

---

## Author Comment (AC1) · 11 Aug 2020

The authors are grateful to the referee for the hard work with manuscript and significant comments. The article is a very large, so a lot of technical information was not included in the text. Here I will try to comment on the main claims: This work was started at 2016, and at that time we used the newest version of WW3. We made sensivity tests with all available parametrization, including the interaction of ice and waves. Model results was compared with several buoy stations in the North Atlantic, the Norwegian Sea, the Barents Sea. But the ST1 scheme was the best choice based on Bias, RMSE and R. The IC0 ice scheme provide the best results. In the Kara Sea the direct wave measurements practically absent, and we use all with is available for us. We can include in the paper the comparison of model results with Saral and Sentinel satellites

directly for the Kara Sea. We can include some sensivity tests with different configuration from 6.07 WW3 but I believe that the storm statistics will not changed. The interaction between ice and waves is a very hard task and needs a separate study. In our paper the focus is on the extreme storm events (with SWH 5-7 m) and this events possible in case wide open water without ice. Thus, the using different ice schemes has no influence on the climate statistics and trends. We agree with all comments on the English language and inaccurate using of terms and will fix it in the next version of manuscript.

---

## Author Comment (AC2) · 11 Aug 2020

The authors are grateful to the referee for the hard work with manuscript and significant comments. The article is a very large, so a lot of technical information was not included in the text.

About Major Comment #1 (It is the same with Referee #1) : This work was started at 2016, and at that time we used the newest version of WW3. We made sensitivity tests with all available parametrization, including the interaction of ice and waves. Model results was compared with several buoy stations in the North Atlantic, the Norwegian Sea, the Barents Sea. But the ST1 scheme was the best choice based on Bias, RMSE and R. In the Kara Sea the direct wave measurements practically absent, and we use

all with is available for us. We can include in the paper the comparison of model results with Saral and Sentinel satellites directly for the Kara Sea. We can include some sensitivity tests with different configuration (ST4, ST6) from 6.07 WW3 but I believe that the storm statistics will not changed.

About Major Comment #2:

A comparison of the different functions with the empirical data showed that the best approximations for the storm recurrence was the Pareto distribution. We have the draft figures of analysis of several function (in attach Gumbel distribution). We have not inserted these estimates into the text because the article is very large. We can include 1 figure with different functions.

About Major Comment #3: We agree with the most part of Referee comments which refers to Grammatical issues. Here I can answer to several disputable comments: Minor Comment #3: L27 - It means that 99% of empirical points lie on the Pareto distribution curve and do not go beyond the confidence interval.

Minor Comment #5: We have tried to put the links to all the most relevant studies regarding the study of wind and waves in the Kara Sea. There are few works directly related to our research, but we decide that more links is better to paper visibility. Minor Comment #7,8,9 - you are absolutely right, but the modelers often use only model documentation and do not read the original source. We will fix it. Minor Comment #11 - this is for a wide range of readers, because for people it is hard to understand the model definition of SWH. Minor Comment #12: "wave heights of 1% and 3% probability of exceedance" - it is like a SWH (which is around 12.5-13% probability), but more extreme, like a Maximum of single waves in package. Wave heights of 1% - it is height of 1 wave from 100 single waves from calculation period (in our case 30 min model step). Minor Comment #13 - we have some problems with open publication of data. It was more stations, but we can not publish pictures. We will clean the manuscript. Minor Comment #15: - yes, 1 time for period 50 years Entire the Kara Sea

[Figure]

Minor Comment #16 - We use only average wave period (WP) Tm02 and mean wave length (WL). We will add the description at chapter 2 Minor Comment #20: If the the wind blows during 2 days - it cause the waves with SWH more than 4 m. It is a one storm event and it is no any links to to 9 hours between storm events.
* * *
[Figure]

[Figure]

**Fig. 1.**

---

## Author Comment (AC3) · 17 Sep 2020

The authors are grateful to the referee for the hard work with manuscript and significant comments. The article is a very large, so a lot of technical information was not included in the text. Here I will try to comment on the main claims: This work was started at 2016, and at that time we used the newest version of WW3. We made sensivity tests with all available parametrization, including the interaction of ice and waves. Model results was compared with several buoy stations in the North Atlantic, the Norwegian Sea, the Barents Sea. But the ST1 scheme was the best choice which based on Bias, RMSE and R. The IC0 ice scheme provide the best results too (but the differences was very small, because measurements was not close to the ice edge) . In the Kara Sea the direct wave measurements practically absent, and we use all wich is available for us.

[Figure]

We can include in the paper the comparison of model results with Saral and Sentinel satellites directly for the Kara Sea. We can include some sensivity tests with different configuration from 6.07 WW3 but I believe that the storm statistics will not changed. The interaction between ice and waves is a very hard task and needs a separate study. In our paper the focus is on the extreme storm events (with SWH 5-7 m) and this events possible generally in case of wide open water without ice. Thus, the using different ice schemes has a little influence on the climate statistics and trends. We agree with all comments on the English language and inaccurate using of terms and will fix it in the next version of manuscript. Next, we analyze the comments on the points. Basically, we agree with everything, but we will comment on several of them. P2L50 data from models are preferable. They are also limited by the presence of sea ice, as waves-in-ice modelling remains quite challenging (e.g Squire, 2020). Also, I would not say that model data are "preferable" to observations. - We mean that the measurements are certainly more accurate and good, but the series are usually short and only at 1-2 points, and this is not enough for analyzing storm activity.

P4L132 In a spectral wave model, SWH is not computed as the average height value of the 33% highest waves. - We understand this, but for untrained readers it is necessary to give a clear definition.